# Specific and Nonspecific Effects of Influenza Vaccines

**DOI:** 10.3390/vaccines12040384

**Published:** 2024-04-05

**Authors:** Nicola Principi, Susanna Esposito

**Affiliations:** 1Università degli Studi di Milano, 20122 Milan, Italy; nicola.principi@unimi.it; 2Pediatric Clinic, Department of Medicine and Surgery, University of Parma, 43126 Parma, Italy

**Keywords:** antibiotics, antimicrobial resistance, autoimmunity, chronic inflammation, influenza vaccination, influenza vaccine

## Abstract

With the introduction of the influenza vaccine in the official immunization schedule of most countries, several data regarding the efficacy, tolerability, and safety of influenza immunization were collected worldwide. Interestingly, together with the confirmation that influenza vaccines are effective in reducing the incidence of influenza virus infection and the incidence and severity of influenza disease, epidemiological data have indicated that influenza immunization could be useful for controlling antimicrobial resistance (AMR) development. Knowledge of the reliability of these findings seems essential for precise quantification of the clinical relevance of influenza immunization. If definitively confirmed, these findings can have a relevant impact on influenza vaccine development and use. Moreover, they can be used to convince even the most recalcitrant health authorities of the need to extend influenza immunization to the entire population. In this narrative review, present knowledge regarding these particular aspects of influenza immunization is discussed. Literature analysis showed that the specific effects of influenza immunization are great enough per se to recommend systematic annual immunization of younger children, old people, and all individuals with severe chronic underlying diseases. Moreover, influenza immunization can significantly contribute to limiting the emergence of antimicrobial resistance. The problem of the possible nonspecific effects of influenza vaccines remains unsolved. The definition of their role as inducers of trained immunity seems essential not only to evaluate how much they play a role in the prevention of infectious diseases but also to evaluate whether they can be used to prevent and treat clinical conditions in which chronic inflammation and autoimmunity play a fundamental pathogenetic role.

## 1. Introduction

Influenza virus infections are very common worldwide [1]. It has been calculated that around a billion cases of seasonal influenza annually occur. Most of them remain asymptomatic or present with mild upper respiratory tract symptoms without fever. In 20% to 40% of cases, influenza presents with the traditionally reported influenza-like illness manifestations, such as fever, sore throat, cough, headache, muscle and joint pain, and severe malaise [2,3,4,5]. Many of these patients require at least one medical visit. Moreover, about 3–5 million cases develop a severe illness that leads to hospitalization and, in some patients, needs admission to the intensive care unit [6]. Among these, between 290,000 and 650,000 cases die. Children younger than 5 years, old people, pregnant women, people with chronic severe underlying diseases including primary or secondary immunodeficiency, and long-term care facility residents are considered at increased risk for influenza complications; those at the highest risk are at risk of hospitalization and death [7].

Due to its high frequency and the relevant number of cases that require medical visits such as hospitalization and yet, despite careful assistance, die, influenza significantly impacts the health system and is associated with a substantial social and economic burden. In the USA, it has been calculated that the direct medical costs of every influenza season that originate from inpatient and outpatient care settings account for USD 3.2 billion. Indirect costs that mainly derive from lost work time and the reduced productivity of patients and caregivers are estimated at USD 8 billion [8,9].

To face these problems, influenza vaccines were developed. After the first inactivated vaccine (IIV) was authorized in the 1940s, several different preparations including inactivated split- and subunit-type and live attenuated vaccines (LAIV) were developed [10]. In some cases, adjuvants were added to improve the immune response [11]. Moreover, innovative tools, such as recombinant technologies and intra-dermal devices, were tested. Initially, preparations potentially effective against three influenza viruses (A/H1N1, A/H3N2, and B) were licensed. More recently quadrivalent vaccines, including a second B virus, were prepared [10]. The exact composition of the influenza vaccine regarding both the type and number of viruses is annually determined by the World Health Organization (WHO, Geneva, Switzerland) [12] based on the most frequent strains isolated in the previous season during continuous surveillance. As of recently, the inclusion of the second B virus was no longer recommended by WHO advisers because in recent years, no confirmed detection of this strain has been documented [13]. With time, the immune response and clinical efficacy of influenza vaccines were progressively improved while maintaining a good safety and tolerability profile. There is evidence that, although with some limitations, mainly in younger children and old people, the use of influenza vaccines significantly reduced the total burden of seasonal influenza and has induced health authorities to progressively extend the list of subjects for whom the vaccine had to be recommended [14]. Initially reserved only for subjects at the highest risk of influenza-related complications, presently, the influenza vaccine is recommended in several countries for all people, regardless of age and health conditions. Even in those countries where universal vaccination is not recommended, younger children and old people are always included in the list [15].

With the introduction of the influenza vaccine in the official immunization schedule of most countries, several data regarding the efficacy, tolerability, and safety of influenza immunization were collected worldwide. Interestingly, together with the confirmation that influenza vaccines are effective in reducing the incidence of influenza virus infection and the incidence and severity of influenza disease, epidemiological data have indicated that influenza immunization could be useful for controlling antimicrobial resistance (AMR) development [16,17]. Moreover, in some studies, results have suggested that influenza vaccines could have nonspecific effects, i.e., induce protection against non-targeted infections and modulate the incidence and course of several immune-mediated diseases [18]. Knowledge of the reliability of these findings seems essential for precise quantification of the clinical relevance of influenza immunization. If definitively confirmed, these findings can have a relevant impact on influenza vaccine development and use. New frameworks for testing, approving, and regulating vaccines capable of collecting data on the overall health of vaccinees during long-term follow-up could be planned. This could allow us to know whether and which influenza vaccine preparations have relevant nonspecific effects, which external factors can influence their clinical importance, and how influenza vaccines should be administered to obtain the greatest cost–benefit balance [19]. Moreover, they can be a relevant additional factor to convince even the most recalcitrant health authorities of the need to extend influenza immunization to the entire population [20]. In this narrative review, present knowledge regarding these particular aspects of influenza immunization is discussed. The MEDLINE (Northfield, IL, USA)/PubMed (Bethesda, MD, USA) database was searched from 1993 to 30 November 2023 to collect the literature. The search included randomized placebo-controlled trials, controlled clinical trials, double-blind, randomized controlled studies, systematic reviews, and meta-analyses. Abstracts were excluded. The following combinations of keywords were used: “influenza vaccine” OR “influenza vaccination” AND “efficacy” OR “effectiveness” OR “effect” OR “antimicrobial resistance” OR “antibiotics”.

## 2. Influenza Vaccine Administration as a Measure to Contain Antimicrobial Resistance Development

In the last decade, AMR to commonly used antibiotics has significantly increased (Figure 1), making bacterial infections harder to treat and increasing the risk of disease spread, severe illness, and death or development of persistent disability.

A comprehensive analysis of the burden of antimicrobial resistance has shown that in 2019, AMR has been associated with 4.95 million deaths among which 1.27 million could be directly attributed to this microbiological problem [21]. As abuse and misuse of antibiotics have been identified as the main cause of AMR development [22,23], reducing and improving antibiotic use through high-quality surveillance and usage guidelines has been considered the most important solution to face AMR problems. Several national governments and scientific institutions, such as the European Union Commission [24], the United States Government [25], and the World Health Organization [26], have initiated or accelerated the development of action plans to combat AMR. Criteria useful to ensure the rational use of antibiotics and reduce AMR have been prepared. Moreover, in many hospitals, especially in industrialized countries, antimicrobial stewardship programs to ensure that these criteria were regularly applied have been implemented [27,28]. However, the results of these initiatives were frequently found to be only partly satisfactory. For some bacteria and in some countries, the percentage of resistant strains has remained high or even increased, leading health authorities to predict that, in the absence of effective measures, in 2050, the annual number of deaths from infections caused by multi-resistant bacteria could reach 10 million, a value significantly higher than that due to cancer or cardiovascular diseases today [29]. Examples in this regard are those collected in the countries of the European Union and the European Economic Area where, from 2017 to 2021, for example, the percentage of methicillin-resistant Staphylococcus aureus and penicillin-resistant Streptococcus pneumoniae was only slightly reduced or increased, respectively. Substantially unchanged or only slightly reduced were the percentage of resistant strains among *Escherichia coli*, *Pseudomonas* spp., and *Klebsiella pneumoniae* [30].

Vaccines, when significantly effective against pathogens that are very common causes of infection and disease, can play a significant role in conditioning AMR emergence [10]. On the other hand, vaccination to reduce AMR is seen as an important global public health issue by WHO, although not all national AMR plans include immunization programs in their AMR plans [31]. Regarding viral vaccines, the reduction in the number of infections due to the virus against which the vaccine had been prepared translates into a parallel reduction in the number of antibiotic prescriptions. The misuse of antibiotics that frequently occurs in patients with viral infection is reduced [10,31]. Moreover, superimposed bacterial infections are reduced, leading to a further limitation in antibiotic use [32,33].

Influenza vaccine characteristics suggest that its use can be very effective in this regard. As previously mentioned, influenza is very common [1]. Influenza vaccine is always significantly effective in reducing influenza virus infections and disease although absolute effectiveness can vary according to several factors, such as the strain matching, type of vaccine and time of administration, and age and health of the vaccinee [10]. Moreover, some clinical studies seem to indicate that in people receiving influenza immunization, antibiotic prescriptions in the months following vaccination are significantly reduced. Details on influenza vaccine effectiveness are summarized in a recent systematic review and network meta-analysis of the placebo- or no vaccination-controlled in head-to-head randomized clinical trials (RCT) published through December 15th, 2020 [34]. A total of 220 RCTs including 100,677 children (<18 years), 329,127 adults (18–60 years), and the elderly (≥61 years)] were included. All vaccines cumulatively achieved major reductions in the incidence of laboratory-confirmed influenza, despite differences according to the previously reported factors. In children, live-attenuated vaccine (LAIV) and inactivated vaccine (IIV) adjuvanted with MF59/AS03 were more efficacious than the inactivated vaccine (IIV) in reducing the risk of laboratory-confirmed influenza virus infection. Compared with 3-IIV, the relative risk (RR) of infection with 3-LAIV and IIV adjuvanted with MF59/AS03 were 0.52 (95% credible interval [CrI] 0.32–0.82) and 0.23 (95% CrI 0.06–0.87), respectively. In adults and the elderly, all vaccines, except the trivalent inactivated intradermal vaccine (3-IIV ID), were more effective than the placebo. RR varied from 0.33 (95% CrI 0.21–0.55) for 3-IVV high-dose (3-IIV HD) and 0.56 (95% CrI 0.41–0.74) for 3-LAIV. However, in the elderly, vaccine efficacy was less pronounced as a significant difference compared to the placebo, which was reached only when the recombinant IV was used [34].

A reduction in antibiotic prescriptions in individuals given influenza vaccine is clearly evidenced in a recent systematic review and meta-analysis of 26 studies [35]. Unfortunately, the quality of the 19 observational studies was generally poor and, despite being in favor of the influenza vaccine, the results of these studies cannot be used to draw definitive conclusions. However, the results of 17 well-conducted RCTs seem adequate to indicate that influenza vaccine use is associated with both the reduction in the proportion of people receiving antibiotics (RR 0.63, 95% confidence interval [CI] 0.51–0.79) and the reduction in number of antimicrobial prescriptions or days of antibiotic use (RR 0.71, 95% CI 0.62–0.83). Moreover, there are data that seem to confirm that influenza vaccines can reduce the risk of superimposed bacterial infections and, consequently, the number of antibiotic prescriptions. A good example in this regard is given by the results of a study in which the incidence of acute otitis media (AOM) in children with or without influenza immunization was measured [36]. AOM is frequently preceded by a viral upper respiratory infection and a superimposed bacterial infection is the cause of AOM in most of the cases [37,38]. Consequently, evaluation of its incidence in patients with and without influenza immunization can be considered a useful test to evaluate the role of influenza vaccine in reducing superimposed bacterial infections. Marchisio et al. performed a prospective randomized single-blinded placebo-controlled study, enrolling 180 children aged 1 to 5 years previously unvaccinated against influenza and with a history of recurrent AOM [39]. These patients were randomized to receive 3-IIV (*n* = 90) or no treatment (*n* = 90) and AOM-related morbidity was monitored every 4 to 6 weeks for 6 months. Among the vaccinees, the number of children experiencing at least one AOM episode during the study period was significantly smaller than in the control group (49; 54.4% vs. 74; 82.2%; *p* < 0.001). Lower than among controls were also the mean number of AOM episodes (0.94 ± 1.12 vs. 2.08 ± 1.52; *p* = 0.03), the mean number of AOM episodes without perforation (0.39 ± 0.66 vs. 1.32 ± 1.49; *p* < 0.001), and the mean number of antibiotic courses (1.47± 1.26 vs. 2.59 ± 1.72; *p* < 0.001). Only in children with a history of recurrent tympanic perforation was the influenza vaccine not effective, suggesting that this kind of AOM is a particular form of AOM requiring a specific preventive and therapeutic approach [39].

Starting from these and other similar findings, the use of vaccines to prevent AMR has been suggested by several experts. The WHO has included vaccines among the measures useful for the prevention and control of AMR [38]. Moreover, the WHO has stressed the need to expand and share knowledge and awareness about the potential role of vaccines in AMR reduction and has highlighted the importance of the rapid development of effective vaccines against those bacteria that already present high levels of AMR [40].

## 3. Mechanisms and Examples of Nonspecific Effects of Vaccines

Several epidemiological studies have shown strong statistical associations between the administration of some vaccines and the development of nonspecific effects, i.e., effects that go beyond the specific protective effects against the targeted diseases and involve unrelated pathogens or immune-mediated diseases [41]. Table 1 summarizes the main examples of nonspecific effects of vaccines different from those against influenza.

These findings were initially interpreted as context-dependent, possibly due to differences in vaccine strains [42] or to interactions with other vaccines [43]. However, immunological studies have shown that nonspecific effects were mainly due to a previously unknown mechanism that is the development of innate immune memory, also named trained immunity, although a role is supposed to be played by the heterologous T-cell immunity also. Trained immunity is a condition for which innate immune system cells, such as myeloid cells (i.e., monocytes, macrophages, and dendritic cells) or lymphoid cells (i.e., natural killer cells and innate lymphoid cells), after the first response to an infectious stimulus that is specifically effective against that pathogen, become able to mount a faster and stronger response when they are again exposed to the same antigen or to unknown new antigens [44]. The heterologous T-cell immunity is, on the contrary, a particular aspect of adaptive immunity for which exposure to a pathogen can result in the activation and expansion of T cells capable of recognizing not only the specific antigen but also different unrelated antigens [45].

Trained immunity can have significant beneficial effects or, on the contrary, enhance vulnerability to diseases not related to the vaccine, particularly in females [46]. In general, live vaccines limit the target infection and, at the same time, exert favorable nonspecific effects providing broad cross-protection against any type of reinfection and protecting the host from the development of some autoimmune diseases. On the contrary, non-live vaccines, despite conferring high protection against the target disease, have unfavorable effects, making people generally more susceptible to infections that are not the target and driving or exacerbating inflammatory or autoimmune responses. Details of the biological mechanisms that are the base for trained immunity development are reported in some recent reviews [47,48,49]. To summarize the present knowledge, it can be concluded that trained immunity involves epigenetic and metabolic reprogramming of the innate immune cells. When this has occurred and the innate immune system is exposed to subsequent time-delayed heterologous stimulations, the adjusted immune responses lead to extended protection against a large number of infectious agents and reduce the risk of chronic inflammation and autoimmune disease development and progression. Unfortunately, misguided trained immunity responses can occur and cause opposite results, with the development of either a chronic hyperinflammatory state or a persistent state of immunological tolerance, a condition in which the activity of the immune system is decreased and the risk of infections or autoimmune diseases is increased.

Live vaccines with documented positive nonspecific effects are the Bacillus Calmette-Guerin (BCG) vaccine, the measles vaccine, the smallpox vaccine, and the live poliovirus (OPV) vaccine. For all these vaccines, several RCTs have clearly indicated that, together with a significant reduction in the target disease, they can reduce all-cause mortality and hospitalizations [50]. A meta-analysis of three trials carried out in Guinea-Bissau enrolling low-birth-weight neonates who had been given BCG at birth has shown that early BCG administration was associated with 38% (RR 0.62, 95% CI 0.46–0.83) and 16% (RR 0.84, 95% CI 0.71–1.00) reduced mortality within the neonatal period or by age of 12 months, respectively [51]. Moreover, BCG vaccine administration may induce protection against malignancies, allergies, and autoimmune diseases, including type 1 diabetes [52]. Regarding the measles vaccine, it has been reported that the introduction of this vaccine in underdeveloped nations had led to a reduction in global pediatric mortality significantly greater than that expected on the basis of vaccine efficacy (30% compared to 10%) [53]. Moreover, the mortality of measles-vaccinated children was lower than that of nonimmunized subjects [54], with girls showing the greatest benefit [55]. Similar findings were reported in the countries where the smallpox vaccine was administered to a large part of the pediatric population. Finally, interesting results were collected when the impact of OPV was studied. OPV administration was associated with a reduction in gastrointestinal infections in Latin America [56], respiratory infections in Russia [57], and global child mortality in several underdeveloped countries [58,59,60].

The negative effect of trained immunity following non-live vaccine administration seems clearly evidenced by the findings of the epidemiological studies evaluating the impact of the diphtheria–tetanus–pertussis (DTP) vaccine. A global analysis of the data collected with studies at low risk of bias seems to indicate that, in low-income countries, the DTP vaccine can cause more deaths from other diseases than it prevents from the target infections. Compared to unvaccinated children, those given DTP have a risk of death five times higher. Risk is higher in females than in males; in children with multiple vaccinations, it depends on the schedule used for immunization [61,62,63]. Deleterious effects occur when DTP is given after live vaccines and in this case, the risk of mortality is doubled. On the contrary, by giving BCG and DTP at the same time, all-cause mortality could be reduced by about 48% [64].

Further support of the hypothesis that non-live vaccines can cause detrimental nonspecific effects is given by the studies regarding inactivated polio vaccine (IPV), malaria vaccine, and hepatitis B vaccine (HBV). IPV has been found to increase all-cause mortality by 10% [65]. Consequently, considering the positive effects of OPV, doubts about the proposed universal change from OPV to IPV to reduce the risk of paralytic paralysis have been raised. It has been calculated that approximately 4000 deaths for each case of vaccine-associated paralytic poliomyelitis may occur, causing a total of more than 300,000 additional deaths each year worldwide [66]. Regarding the malaria vaccine, it has been reported that administration of the RTS,S/AS01 vaccine, although modestly effective in children against the target disease, was accompanied by a significant in all-cause mortality in girls (mortality rate ratio [MRR] 1.91; 95% CI 1.30–2.79; *p* = 0.0006) although not in boys (MMR 0.84; 95% CI 0.61–1.17; *p* = 0.3343) [67]. Data regarding HBV were collected by Garly et al. with a study in which the mortality rate of children who had received measles and hepatitis B vaccines was compared with those of children given only the measles vaccine [68]. Results clearly indicated the negative impact of HBV as children given this vaccine had higher mortality than those without immunization (MRR 1.81; 95% CI 1.19–2.75) with the difference being particularly strong for girls (MRR 2.27; 95% CI 1.31–3.94). As far as 3IIVs and 4IVVs are concerned, however, the potential association with negative nonspecific effects is not defined. For these non-live vaccines, a great number of data are available but conclusions cannot be drawn.

## 4. Influenza Immunization and Nonspecific Effects

Two different types of influenza vaccines are presently available, IIVs and LAIV. IIVs are non-live vaccines, whereas LAIV is a live vaccine [69]. In agreement with what has been demonstrated for other vaccines, IIVs would have been expected to induce negative nonspecific effects and LAIV beneficial effects. Studies regarding LAIV are few but they seem to indicate that the results of its administration correspond to expectations including the induction of positive nonspecific effects. In a study in which 6,569 children were immunized with LAIV, it was found that this vaccine could have significant indirect effects reducing the total number of non-influenza medical-attended respiratory infections in both children and adults [70]. Moreover, a recent experimental study has shown that the administration of X-31ca, a donor virus for the preparation of an LAIV, could provide nonspecific cross-protection against respiratory syncytial virus (RSV). Administration to experimental animals of X-31ca before RSV infection was associated with a significant reduction in RSV replication. This was, in turn, associated with an immediate release of cytokines and infiltration of leukocytes into the respiratory tract suggesting a remodulation of innate immune activity [71].

On the contrary, what is known about IIVs is very different. Contrary to expectations, the available data regarding IIVs do not allow us to confirm or exclude the chance that these vaccines can exert a nonspecific effect. The results of studies specifically planned to evaluate the clinical impact of IIVs are highly conflicting. A great number of studies have reported that IIV has no effect on the incidence of noninfluenza respiratory infections [72,73,74,75,76,77]. Other studies have found negative effects [78,79,80]. Finally, some studies have found that IIVs can exert substantial protective effects. [81,82,83,84,85,86].

Some examples can illustrate these findings. Negative nonspecific effects were reported in some studies carried out in some African countries during the 2009 A/HIN1 2009 influenza pandemic. Children who were given the non-live A/H1N1 pandemic vaccine had higher age-adjusted mortality rates after immunization than children who did not receive the vaccine [87]. Adults immunized with 3IIV including the A/H1N1 2009 pandemic strain had higher rates of respiratory symptoms and absence from work than those without vaccination [88]. In a study in which 115 children were randomized to receive 3IIV or placebo, monitoring over the following 9 months revealed that the risk of virologically confirmed respiratory non-influenza infections, mainly those due to rhinovirus and coxsackie/echovirus, was significantly greater among vaccinees than among controls (RR 4.40; 95% 1.31–14.8) [80]. No effect was evidenced by Skowronski et al. who assessed influenza vaccine effectiveness against influenza and non-influenza respiratory viruses using historic datasets of the community-based Canadian Sentinel Practitioner Surveillance Network, spanning 2010–2011 to 2016–2017 [77]. A total of 4281 influenza, 2565 non-influenza respiratory infections, and 3841 pan-negative cases were enrolled. IVV was found to be effective against influenza cases (45%) but it did not show any efficacy against non-influenza respiratory viruses. Finally, a good example of a favorable nonspecific effect of IIV is given by the data collected by Debisarun et al. [81]. These authors compared the incidence of laboratory-confirmed COVID-19 cases among 4IIV vaccinated and unvaccinated employees of a University Medical Center in the Netherlands during the first two waves of the pandemic (March–June 2020 and November 2020–January 2021, respectively). Results evidenced a significant protective effect of 4IIV against COVID-19 as people receiving this vaccine had 37% and 49% lower risk of SARS-CoV-2 infection in the first and the second pandemic waves, respectively. During the first period, 107 of the 3201 (3.34%) individuals who were not vaccinated against influenza had COVID-19, compared to 77 of 3655 (2.11%) of those who were given the vaccine (RR 0.63, 95% CI 0.47–0.84, *p* = 0.0016). In the second wave, even greater evidence of the protective effect of 4IIV was shown. Among 6370 individuals without 4IIV, 250 COVID-19 cases (3.92%) were diagnosed. On the contrary, among 4529 vaccinees, only 91 COVID-19 cases (2.00%) were found (RR 0.51, 95% CI 0.40–0.65; *p* < 0.0001) [81]. Similar positive findings were reported in children regarding the impact of RSV infections. Analysis of RSV infection incidence during a 5-year period in Australia revealed that receipt of 3IIV was associated with a relevant reduction in RSV hospitalizations, especially in those <2 years. Rate reduction was 2.27 per 1000 (95% CI −3.26, −1.28) in these subjects and 0.53 per 1000 (95% CI −1.04, −0.02) in those 2–7 years [89].

Together with the impact of nonspecific infections, several lines of data seem to suggest that influenza vaccine administration may be protective in the development and/or progression of a variety of chronic diseases [90]. The most compelling data come from studies carried out in patients with cardiovascular diseases such as coronary heart disease, heart failure, and stroke. In most cases, results seem to indicate that patients receiving the influenza vaccine are at lower risk of developing or worsening these cardiovascular problems than patients without vaccination [91]. A recently published systematic review and meta-analysis in which 5 RCTs with very low risk of bias strongly supports this conclusion [92]. A total of 9059 adult patients with well-defined cardiovascular diseases were enrolled. Of them, 4529 had received 3IIV and 4530 a placebo. In the 9 months following immunization, a major cardiovascular event occurred in 517 vaccinees and in 621 controls, clearly suggesting a relevant protective effect of the influenza vaccine (RR 0.70, 95% CI 0.55–0.91). The stratified analysis confirmed this finding, showing a significant impact of vaccination on the risk of myocardial infarction development (RR 0.74, 95% CI 0.56–0.97) and of cardiovascular death events (RR 0.67, 95% CI 0.45–0.98). No effect on the risk of stroke was, however, found. Moreover, some studies have suggested a potential protective effect of influenza vaccine in the development and progression of type 1 diabetes (T1D) [92,93], cancer [94,95,96,97,98], and Alzheimer’s disease [99,100,101].

Partially discordant are, on the other hand, the studies aimed at evaluating the epigenetic and transcriptional reprogramming as well as cytokine responses of immune cells after IIV administration. Debisarun et al. reported that 4IIV was able to modify innate immune cell activity with a reduction in systemic inflammation and modulation of the transcriptional program and cytokine production upon stimulation with the SARS-CoV-2 virus [81]. A lower antiviral response was, on the contrary, reported by Wimmerset al. who, despite confirming that vaccination stimulates persistent epigenomic remodeling of the innate immune system, found a persistent impaired cytokine response to stimulation [101].

Table 2 summarizes the nonspecific effects of influenza immunization.

Such different results make it impossible to draw any conclusions and leave the problem of the development of nonspecific effects of IIVs totally unsolved. Despite other factors, such as characteristics of the study population, vaccine formulations, or outcome measurements, which may explain the different results, it seems highly likely that the most important causes of this finding are the methodological limitations of most studies. The greatest part of the available data has been collected with observational studies that frequently have several methodological limitations that can lead to debatable and contrasting results. Information on confounding factors and effect modifiers can be missing and correction for confounders is sometimes not possible, causing over- or underestimation of outcomes. Moreover, the selection of subjects to enroll can be a limit per se as individuals willing to be vaccinated against influenza may also be those more likely to respect the personal protection rules against all the other respiratory infections, causing a potential overestimation of the positive effect of IIVs. Finally, it cannot be excluded that a role may be played by the characteristics of the IIVs themselves. The inclusion of adjuvants may be critical for the induction of different immune responses and the development of nonspecific effects. Further studies, mainly RCTs, are needed to solve the problem and establish whether and which nonspecific effects can have IIVs. Attempts in this regard have already been made although no results have been published until now. Regarding the impact on T1D, a trial (NCT05585983) testing the hypothesis that influenza vaccination can be more effective than placebo in sustaining β cell function in early T1D has been initiated. Regarding cancer, two studies have been planned. The first intends to evaluate the safety and efficacy of treating patients with early colorectal cancer with intratumoral influenza vaccine as a downstaging and immune response enhancing treatment prior to intended curative surgery (NCT04591379). The second study intends to evaluate the efficacy of influenza vaccination together with perioperative Tadalafil for decreasing the chances of the spread of disease post-surgery in patients suffering from a primary abdominal malignancy (NCT02998736). It is essential to substantiate the potential ability of IIVs to induce positive nonspecific effects; however, clinical studies are accompanied by a careful evaluation of the ability of IIVs to induce trained immunity. The example of the study by Debisarun et al. is paradigmatic in this regard [81]. These authors were able to demonstrate the clinical efficacy of 4IIV in reducing the risk of COVID-19, which was accompanied by improved responsiveness of innate immune cells to heterologous viral stimuli, suggesting the activation of a sustained trained immunity [81]. When several studies report a strict association between the development of positive nonspecific effects and the induction of trained immunity, the evidence for the potential use of IIVs to reduce infections and limit autoimmunity or choric inflammatory disease will be definitively ascertained.

## 5. Conclusions

The specific effects of influenza immunization are great enough per se to recommend systematic annual immunization of younger children, old people, and all individuals with severe chronic underlying diseases. Moreover, the evidence that influenza vaccine administration to healthy adolescents and adults reduces influenza virus circulation with relevant benefits from a medical, social, and economic point of view is a further factor for very large use of this preventive measure.

However, the benefits of influenza immunization are not limited to the reduction in the total burden of influenza infection. Several RCTs have clearly evidenced that influenza immunization can significantly contribute to limiting the emergence of AMR, a problem that already causes a significant increase in the risk of death from bacterial infectious diseases and which is expected to quickly become the most important cause of mortality. This further supports the largest use of influenza vaccines and explains why several national health authorities as well as the WHO have stressed the importance of this particular aspect of influenza vaccine activity and strongly sponsored the development of vaccines effective against the pathogens for which AMR is clinically relevant.

On the contrary, the problem of the possible nonspecific effects of influenza vaccines remains completely unsolved. The data relating to LAIV, although apparently favorable, are too limited to draw definitive conclusions and to suggest the preferential use of these vaccines instead of IIVs. Regarding the latter, despite the great number of studies, available data do not allow us to definitively decide whether these vaccines have nonspecific effects. Some clinical and in vitro evaluations seem to suggest that these vaccines may be an exception to what has been demonstrated for many other non-live vaccines. Unlike these, IIVs may allow the development of a substantial trade immunity leading to nonspecific cross-protection against infections, a reduction in chronic inflammation, and the development of autoimmunity. However, many studies have opposite results and most of those showing favorable data have relevant methodological limitations that make results debatable. A large series of RCTs associated with well-conducted immunological studies are needed to definitively solve the nonspecific effects of both LAIV and IIVs. The definition of their role as inducers of trained immunity seems essential not only to evaluate how much they play a role in the prevention of infectious diseases but also to evaluate whether they can be used to prevent and treat clinical conditions in which chronic inflammation and autoimmunity play a fundamental pathogenetic role, significantly extending their original role. Further studies are needed to address the possible molecular or immunological mechanisms that lead to positive and negative effects of influenza vaccination on different populations.

## Figures and Tables

**Figure 1 vaccines-12-00384-f001:**
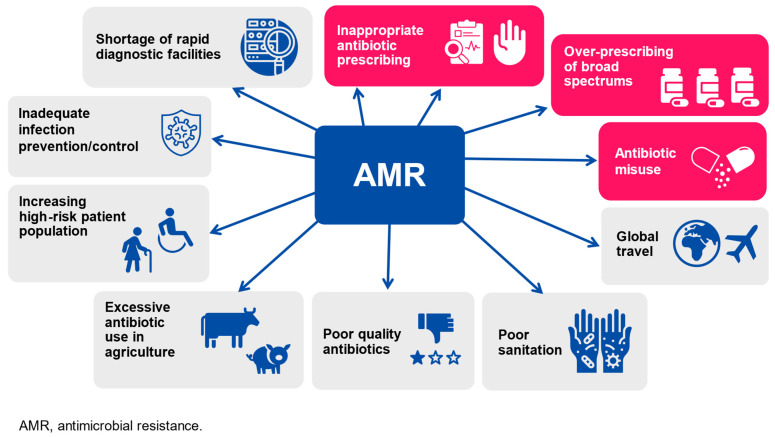
Major drivers of the development and spread of antimicrobial resistance (in red the main causes of antimicrobial resistance development).

**Table 1 vaccines-12-00384-t001:** Main examples of nonspecific effects of vaccines differ from those against influenza.

Type of Effect and Vaccine	Nonspecific Effects
Positive	
Bacillus Calmette-Guerin (BCG) vaccine	Reduced mortality within the neonatal period or by age of 12 months; protection against malignancies, allergy, and autoimmune diseases, including type 1 diabetes
Measles vaccine	Reduction in global pediatric mortality, with girls showing the greatest benefit
Smallpox	Reduction in global pediatric mortality
Live poliovirus vaccine (OPV)	Reduction in gastrointestinal infections in Latin America, of respiratory infections in Russia, and of global child mortality in several underdeveloped countries
Negative	
Diphtheria-tetanus-pertussis (DTP) vaccine	Increased deaths from other diseases than it prevents from the target infections when is given after live vaccines
Inactivated polio vaccine (IPV)	Increase all-cause mortality by 10%
Malaria vaccine RTS,S/AS01	Increase in all-cause mortality in girls
Hepatitis B vaccine (HBV)	Increase in mortality with the difference being particularly strong for girls

**Table 2 vaccines-12-00384-t002:** Main examples of nonspecific effects of influenza vaccines influenza.

Type of Vaccine	Nonspecific Effects
Live attenuated influenza vaccine (LAIV)	Some data on the reduction in the total number of non-influenza medical attended respiratory infections in both children and adults
	Nonspecific cross-protection against respiratory syncytial virus, suggesting a remodulation of innate immune activity
Inactivated influenza vaccine (IIV)	Conflicting results on the incidence of non-influenza respiratory infections
	Significant protective effect of 4IIV against COVID-19
	Reduction in RSV hospitalizations in children, especially in those <2 years, with 4IIVs
	Reduced risk of developing or worsening coronary heart disease and heart failure; no effect on stroke
	Potential protective effect of influenza vaccine in development and progression of type 1 diabetes (T1D), cancer, and Alzheimer’s disease
	Conflicting results on epigenetic and transcriptional reprogramming as well as cytokine responses of immune cells after administration

## Data Availability

Not applicable.

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
