# Peer review of "Specific and Nonspecific Effects of Influenza Vaccines"

_vaccines, 2024, doi:10.3390/vaccines12040384_

Round 1

Reviewer 1 Report

Comments and Suggestions for Authors

Profound an well-structured overview on indirect and non-specific effects of inluenza- (and other) vaccines.

Minor comments: 
Table 1: The association of the lines on the left and right-hand side are not so obvious.
Table 2: The page break is in the middle of the table.

Comments on the Quality of English Language

There are sporadic mistakes, but they are not relevant to the understanding of the text.

Author Response

Thank you very much for your comments. We revised the manuscript accordingly.

Table 1 has been modified in order to improve readability.

Table 2 has been revised as suggested.

Reviewer 2 Report

Comments and Suggestions for Authors

This work by Principi and Esposito provides important insights into the broad impacts of influenza vaccination, including the role in combating antimicrobial resistance (AMR) and potential non-specific effects.

Below are some suggestions to be considered before publication. I tend to view them as minor comments since they can all be addressed by the authors without the need to re-review the literature. 

General comments

- The distinction between specific and non-specific effects, especially in the context of trained immunity, is key for the purposes of this review. I recommend adding an initial section that clearly defines these concepts to provide a solid foundation for the review.

- The authors highlighted the conflicting results regarding the non-specific effects of influenza vaccines. A deeper analysis exploring the reasons behind these discrepancies could benefit a Vaccine reader. Factors such as study populations, vaccine formulations, or outcome measurements could be influencing these results. 

- Expanding on how the findings could impact public health policy and vaccination strategies would make the review more applicable to a broader audience. 

Specific comments: 

- Paragraphs Lines 74-78 and 80-82: These sections feel somewhat vague and potentially misleading regarding authorities' stance on vaccine effectiveness. I suggest rephrasing to emphasize the advantages of extending influenza immunization to the entire population, especially in terms of the cost-benefit balance.

-  I recommend including and discussing the study found at https://pubmed.ncbi.nlm.nih.gov/36253789/. This reference could provide substantial support to your arguments regarding the influenza vaccine's role in the fight against AMR.

- Lines 152-164:  I suggest summarizing the most important points from studies, making your review easier to read, without delving that much into single studies. 

Lines 228-276: The detailed discussion on other vaccines and trained immunity, though informative, feels somewhat off-topic. A more focused explanation relevant to influenza vaccines would maintain the review's thematic coherence.

- If Table 1 is derived from a single study (referenced as n.39), its necessity should be reconsidered. Instead, summarizing key points in the narrative might be more effective. Please clarify. 

- Lines 298-300: The mention of influenza vaccines' effects on COVID-19 feels somewhat tangential and underdeveloped. Given the retrospective nature of supporting studies and potential biases, I suggest either expanding this discussion to address these nuances or considering the removal of this statement.

Author Response

The distinction between specific and nonspecific effects of vaccines is reported in several sections of the paper, including the introduction.

The possibility that the different results of studies regarding nonspecific effects of influenza vaccines could be ascribed to several factors, including study population characteristics, vaccine formulations or outcome methods of evaluation has been reported in the paper.

It has been detailed how data confirming the clinical relevance of influenza vaccine nonspecific effects may modify traditional evaluation of vaccine efficacy and safety  and increase vaccine use by the general population.

Advantages of universal influenza vaccination have been reported.

The suggested reference has been included in the text.

Lines 152-164. This section reports the most recent data showing why influenza vaccine is per se important for influenza infection and disease prevention. The importance of the vaccine for its specific effect is the base for discussion of nonspecific effects. We do not think that it should be reduced or deleted.

Lines 228-276. This section has been included to inform the reader that vaccines can have nonspecific positive or negative effects and to make it easier to understand potential differences between live and inactivated influenza vaccines

Table 1 has been modified.

Lines 298-300. As suggested, the sentence has been modified.

Reviewer 3 Report

Comments and Suggestions for Authors

The article is very interesting in addressing the effects of vaccination on the population, both positive and negative. Data collection supports the conclusions of the work. It would only be advisable to address, in a general way, the possible molecular or immunological mechanisms that lead to these positive or negative effects. If they are not mentioned in the reviewed studies, then address the importance of studying them.

Author Response

Thank you for your positive evaluation. We clarified that further studies are needed to address the possible molecular or immunological mechanisms that lead to positive and negative effects of influenza vaccination on different populations.

Reviewer 4 Report

Comments and Suggestions for Authors

The manuscript is well written and easy to read.

This quit ambitious review covers different aspects related to many different vaccines. The main conclusions are focused on influenza vaccines and on their direct effect and the effect in reduction of antimicrobial resistance. Both topics are not original since have been reviewed in recent studies. This manuscript also reviews studies that suggested non-specific effects of different vaccines, but these are not concluding and sometimes contradictory results.  Messages from this last part are speculative.

Since vaccines activate immunological mechanisms, including the adaptive immunity, some non-specific effects are possible. Many different effects are possible, but most of them have not been evaluated. It is difficult to conclude about the balance between positive and detrimental effects. Most of reported non-specific effects are non-demonstrated hypothesis. Other nonspecific effects have been observed only in specific areas and they may not be valid for other countries. Results from observational studies are prone to bias that can explain some of the associations mentioned in this review. Publication bias tends to highlight studies that found associations over other studies than rule out these associations. The conclusions of this revision would have to be hypothesis for study instead of public health recommendations.

To conclude about non-specific effects of influenza vaccines by immunological mechanisms may be risky since possible effects may have opposite directions.

The review of influenza studies are based on evaluations of one season vaccination, while programs stated annual vaccination. Additional studies may be necessary to demonstrate the effects in a strategy of routine annual vaccination.

This review may be biased, because highlight studies that found association between vaccination and non-specific effects, but omit many other studies that ruled out similar associations.

Specific comments

Line 15 and 80. Consider if this sentence is scientifically appropriate “Moreover, they can be used to convince even the most recalcitrant health authorities of the need for extending influenza immunization to entire population.” This sentence seems somewhat speculative and states non-demonstrated assumptions. Furthermore, public health recommendations or decisions should incorporate other relevant points as quantification of the effect, economic evaluation, etc.

Line 55. Quadrivalent vaccines have no longer recommended by WHO. Consider to change this sentence.

Table 1. The listed non-specific effects of vaccines are non-demonstrated hypothesis or statistical associations that have not rule out possible bias. Consider if you are arising non-relevant and false hypothesis.

Author Response

We totally agree with the comments of the reviewer. In the conclusions, limitations of the presently available studies and need for further clinical and immunological studies are highlighted.

Line 15 and 80. These sentences have been modified according to the suggestion of this and another reviewer.

Table 1. It has been added in the text that many data reported in the literature regarding nonspecific effects of vaccines are derived from non-demonstrated hypothesis or simple statistical associations.

Reviewer 5 Report

Comments and Suggestions for Authors

The paper reviewed presents an issue of the utmost relevance  if it can be proven that  vaccines can confer  heterologous protection. In particular, it deals with Influenza vaccines. It is well structured and presented in an orderly manner.    These findings provide evidence to underscore the importance of  universal Influenza vaccination.

Author Response

Thank you very much for your positive evaluation. The manuscript has ben revised according to comments received from the other reviewers.